# Shear Stress-Induced AMP-Activated Protein Kinase Modulation in Endothelial Cells: Its Role in Metabolic Adaptions and Cardiovascular Disease

**DOI:** 10.3390/ijms25116047

**Published:** 2024-05-31

**Authors:** Philipp C. Hauger, Peter L. Hordijk

**Affiliations:** Department of Physiology, Amsterdam UMC, Amsterdam Cardiovascular Sciences, Microcirculation, De Boelelaan 1117, 1081 HV Amsterdam, The Netherlands; p.c.hauger@amsterdamumc.nl

**Keywords:** endothelial cells, vasculature, AMPK, fluid shear stress, mechanosensation, metabolism, cardiovascular disease

## Abstract

Endothelial cells (ECs) line the inner surface of all blood vessels and form a barrier that facilitates the controlled transfer of nutrients and oxygen from the circulatory system to surrounding tissues. Exposed to both laminar and turbulent blood flow, ECs are continuously subject to differential mechanical stimulation. It has been well established that the shear stress associated with laminar flow (LF) is atheroprotective, while shear stress in areas with turbulent flow (TF) correlates with EC dysfunction. Moreover, ECs show metabolic adaptions to physiological changes, such as metabolic shifts from quiescence to a proliferative state during angiogenesis. The AMP-activated protein kinase (AMPK) is at the center of these phenomena. AMPK has a central role as a metabolic sensor in several cell types. Moreover, in ECs, AMPK is mechanosensitive, linking mechanosensation with metabolic adaptions. Finally, recent studies indicate that AMPK dysregulation is at the center of cardiovascular disease (CVD) and that pharmacological targeting of AMPK is a promising and novel strategy to treat CVDs such as atherosclerosis or ischemic injury. In this review, we summarize the current knowledge relevant to this topic, with a focus on shear stress-induced AMPK modulation and its consequences for vascular health and disease.

## 1. Introduction

The transport of oxygen, nutrients, and cells through the human circulation to all cells in the human body is a vital physiological process. The circulation comprises blood vessels that are stratified in structure: the innermost layer is the tunica intima, followed by the tunica media, and the outermost tunica adventitia [1]. The intima is made up of a single monolayer of ECs that form a tightly regulated barrier to control the molecular and cellular transport to and from the surrounding tissue [2]. In larger vessels, the media is composed of multiple layers of mural cells, including vascular smooth muscle cells in arteries and veins and pericytes in the wall of microcirculatory blood vessels. The adventitia comprises fibroblasts and extracellular matrix that provide structural support to the vessels, as well as small vessels (vasa vasorum) and nerve endings [3] (graphical summary: Figure 1).

ECs fulfill vital roles in controlling oxygen and nutrient delivery to surrounding tissue and organs, regulating blood flow and vascular tone, as well as immune cell migration and inflammatory responses [4]. In accordance with their role in these important functions, dysregulation of ECs is associated with CVDs such as atherosclerosis [5], hypertension [6], aneurysms [7], and metabolic diseases [8].

Due to the spatial organization at the luminal side of the vascular wall, the ECs are constantly exposed to the forces that are generated by the blood flow, which can be collectively termed hemodynamic forces [9]. These hemodynamic forces exerted on ECs can be further subdivided into two types of mechanical stressors: stretch (cyclic strain) and shear stress [10].

Cyclic strain (CS) on ECs results from the pulsatile nature of blood flow during the cardiac cycle. Due to its location close to the heart, the aorta experiences the highest magnitude of CS, which results in its periodic expansion with each heartbeat. This expansion is a physiologically relevant feature for the aorta to offset the pulsatile forces and equilibrate for steady blood pressure in the periphery. Consequently, increased pathological aortic stiffness poses a cardiovascular risk, as it may cause hypertension and organ damage [11].

Shear stress is a force that is created by the friction of the flowing blood over the endothelium [12]. The magnitude of shear stress τ_ω_ on the vessel wall can furthermore be mathematically determined based on the blood viscosity (μ), the mean volumetric flow rate (*Q*), and the vessel radius *r* and π using the following equation [13]:$$\tau_{\omega }=\frac{4\mu Q}{\pi r^{3}}$$

Consequently, the wall shear stress varies highly across different vascular beds and physiological states. Moreover, based on the geometry of the vessel wall, two types of shear stress occur throughout the vascular system: laminar shear stress (LSS), which is present in more linear regions of larger arteries, capillaries, and veins, is characterized by a relatively constant blood flow. Disturbed shear stress (DSS), on the other hand, is characterized by a turbulent or oscillatory (OF)/disturbed (DF) blood flow, which can be found in curvatures and bifurcations of the vascular network. The Reynolds number (*R_e_*) is a numeric value that defines turbulent flow (when *R_e_* is greater than 2000) and is calculated as follows:$$R_{e}=\frac{\rho u2r}{\mu }$$
where *ρ* represents the blood density, u is the flow velocity, *r* is the radius of the vessel lumen, and μ is the blood viscosity [13].

Numerous studies in the past explored the phenomenon that ECs in areas of LSS show a phenotype that is distinct from that in regions of DSS. In regions of LF and high shear stress, such as in the straight part of the arterial tree, ECs adopt an elongated phenotype that is axially polarized in the direction of flow (planar polarity). However, in areas of low shear stress and TF, ECs do not display these morphological features but instead are rather cobblestone-shaped without being clearly polarized [14].

Arguably, the most prominent observation in that context stems from research in atherosclerosis, where it was established that atherosclerotic lesions predominantly form in regions of DSS, whereas regions of LSS are protected from the development of atherosclerosis [15]. Moreover, ECs in regions characterized by DSS exhibit elevated levels of pro-inflammatory and oxidative stress markers alongside increased expression of proapoptotic and vasoconstrictive markers [16].

These spatial differences suggest that ECs are capable of sensing and reacting to different (levels of) stressors. In fact, ECs have the ability to detect these differences in hemodynamic forces, and it is well established that different types of mechanical stimuli result in specific EC adaptations. This process of sensing and adapting to mechanical stimuli, called mechanosensitivity, is crucial for the maintenance of the endothelial barrier and is a hallmark of EC physiology and vascular health [17].

Besides regulating the previously described cellular stress responses and morphological adaptions, it was also shown that shear stress modulates EC metabolism [18]. Despite their direct exposure to oxygen in the bloodstream, ECs exhibit a relatively low mitochondrial content (2–6%) [19] compared to other cells, such as hepatocytes (28%) [20] or cardiomyocytes (~30%) [21]. It was furthermore shown that ECs depend on anaerobic glycolysis as the primary mechanism for adenosine triphosphate (ATP) production [22,23].

It was previously discovered that ECs exhibit metabolic adaptations to support functional requirements, transitioning from a low glycolytic state in a quiescent monolayer to a heightened glycolytic state during angiogenesis [23]. This is reminiscent of findings in epithelial cells (EpiC) that line the outer layers of the body, such as the skin and gut, which are also mechanosensitive. In EpiCs, the application of mechanical stress results in glucose uptake to fuel energy-demanding actin cytoskeleton rearrangements, a process that is mainly regulated via AMPK [24]. This indicates that metabolic regulation in ECs and EpiCs is similar and that AMPK plays a key role in EC metabolic regulation as well. Indeed, the metabolic adaptations to shear stress in ECs are predominantly governed by AMPK, the same kinase that is responsible for regulating glucose uptake in EpiCs under mechanical stress [25]. However, in contrast to previous findings in EpiCs exposed to mechanical stress, ECs downregulate their glucose uptake upon LSS exposure [18], underscoring that EpiCs and ECs are distinct cell types with common but also distinct signaling and regulatory pathways.

AMPK is a highly conserved kinase that is present in many cell types and is well studied in its primary context as a metabolism-sensing kinase. AMPK is activated when the ATP to AMP (adenosine monophosphate) ratio exceeds a defined threshold, subsequently resulting in the AMPK-mediated enhancement of autophagy and mitophagy. This adaptive response addresses metabolic needs and counters energetic stress [26].

Taken together, in ECs, AMPK functions as a metabolic sensor and a mechanoresponsive element, controlling the upregulation of metabolic adaptive pathways in response to shear-induced mechanical stimulation. This positions AMPK at an intriguing intersection of signaling pathways, facilitating the integration of metabolic responses with mechanosensation in ECs.

This review aims to provide a thorough overview of current insights into the role of AMPK in EC mechanosensitivity and its contributions to metabolic adaptations. The initial sections will elucidate the structural aspects and regulatory mechanisms governing AMPK. Subsequently, we will delineate the contemporary understanding of metabolic adaptions in response to shear stress mediated by AMPK. Finally, we will provide a comprehensive overview of AMPK’s involvement in CVD and its potential as a therapeutic target.

## 2. Structure and Activation of AMPK

AMPK is conserved across eukaryotic cells and was identified in protists, fungi, plants, and animals [27,28]. It is widely acknowledged as a central sensor for the metabolic state of the cell. AMPK has a heterotrimeric structure formed by catalytic α subunits and regulatory β and γ subunits [29]. Multiple isoforms of genes encoding these subunits have been identified, and in humans, two isoforms of the α-subunit (α1 and α2) are expressed, encoded by the genes PRKAA1 and PRKAA2 [30], as well as two β-subunit isoforms (β1 and β2, encoded by the genes PRKAB1 and PRKAB2) [31] and three γ subunit isoforms (γ1, γ2 and γ3, encoded by PRKAG1, PRKAG3, and PRKAG3) [32].

As a result of possible combinations of subunits, AMPK can theoretically form 12 distinct complexes in humans. The full functional and regulatory distinctions between these complexes remain unclear to date [26]. However, certain tissues and cell types have been associated with the expression of a specific AMPK complex, indicative of functional differences and tissue specialization [33], which we will further address in the next paragraph in the context of ECs.

AMPK, serving as a metabolic sensor, undergoes activation in response to diverse physiological stimuli associated with metabolic stress, such as nutrient deprivation, exercise, and cell proliferation. The canonical activation of AMPK acts via sensing ATP to AMP and ADP (adenosine diphosphate) ratios, which are shifted during metabolic stress towards higher ADP and AMP levels. Under such conditions, AMP or ADP binds to the γ subunit of AMPK, resulting in its activation via modulating the phosphorylation level of Thr172 on the α-subunit by allosteric activation, increasing the phosphorylation at Thr172 while inhibiting dephosphorylation at the same site [34]. Furthermore, AMPK kinases (AMPKKs) have been identified, most prominently calcium/calmodulin-dependent protein kinase kinase (CAMKK) [35] and liver kinase B1 (LKB1) [36], that directly modulate the phosphorylation levels of the α-subunit in response to metabolic stress [37]. In addition, noncanonical AMPK-activating pathways exist (reviewed by Steinberg et al. [34]), for example, via transforming growth factor-β-activated kinase 1 (TAK1). Interestingly, a study delineating the consequences of LKB1 deletion in cardiac and skeletal muscle cells unveiled additional specificity in AMPK activation. This work suggests that LKB1 predominantly controls the AMPK α2 subunit, indicating that an alternative mechanism regulates the α1 subunit [38].

Metabolic activation of AMPK has a multitude of downstream effects, modulating fatty acid (FA) [39], cholesterol [40], and carbohydrate metabolism [41], as well as protein synthesis [42] and autophagy via mTOR (mammalian target of rapamycin) and ULK1 (unc-51-like kinase 1) [43]. In sum, the activation of AMPK, as well as its downstream effects, are complex and diverse, positioning AMPK at the center of a variety of cellular pathways. This review specifically explores the regulation of AMPK by shear stress and its consequences in ECs.

### AMPK in Human Endothelial Cells

It was shown in human umbilical vein endothelial cells (HUVECs) that the predominant AMPK complex is made up of the α1, β1, and γ1 subunits [44]. Interestingly, further studies into the subcellular localization and activity of AMPK subunits in ECs revealed that a specific AMPK complex made up of α2, β2, and γ2 subunits is found in nuclear fractions of HUVECs but not in cytosolic or cytoskeletal fractions. Functionally, this specific α2β2γ2 complex was linked to mitosis in HUVECs, connecting specific AMPK subunit activation to distinct cellular processes [37] and supporting the hypothesis that in ECs as well, distinct isoforms exhibit specific regulatory functions.

As described in the previous section, the role of AMPK as a metabolic sensor has been well established for several cell types. However, the role of AMPK in ECs is more complex, with several stimuli activating AMPK in ECs. Besides the canonical stimuli such as elevated AMP and ADP levels and a shift in metabolic needs, AMPK was repeatedly shown to be activated by LSS in ECs [18,45,46], albeit the exact mechanism driving this response remains to be elucidated. Recent findings identified protease-activated receptor-1 (PAR-1) as a mechanosensitive element upstream of AMPK activation [47]. In response to LF, PAR-1 was found to be internalized in endosomes in HUVECs, and PAR-1 depletion resulted in phosphorylation of Src, ERK5, HDAC5, and AMPK, as well as EC alignment in the direction of flow, stress fiber formation, and an increase in anti-inflammatory and atheroprotective gene expression [47]. However, due to the variety of distinct shear stress-induced AMPK-mediated responses, it is unlikely that PAR-1 is the sole shear sensor in ECs controlling AMPK, and additional research is necessary to increase our current knowledge on this topic.

## 3. Effects of Shear-Mediated AMPK Activation in ECs

### 3.1. AMPK-Dependent Metabolic Adaptions in Response to Shear Stress

Although directly exposed to oxygen in the blood, ECs rely predominantly on glycolysis for ATP generation, next to FA and amino acid metabolism. Additionally, ECs have been observed to transition from a metabolically low, quiescent state, typical of monolayer culture, to a state of increased glycolysis upon exposure to factors such as vascular endothelial growth factor (VEGF), for example, during angiogenesis [48]. Primarily, this metabolic adaption follows an increased activity of phosphofructokinase-2/fructose-2,6-bisphosphatase 3 (PFKFB3), a rate-limiting enzyme for glycolysis [23]. ECs furthermore adjust their glycolytic rate in reaction to shear stress, with unique alterations correlating with LF, distinct from adaptations observed following exposure to oscillatory flow (OF) [49]. As a mechanosensitive, metabolism-regulating kinase, AMPK is thus at the center of EC metabolic adaptions in response to flow.

### 3.2. AMPK-Mediated Regulation of Glycolysis

An intriguing aspect of EC metabolism in response to flow is that ECs downregulate glycolysis when exposed to LSS, independent of AMPK via Krüppel-like factor 2 (KLF2)-mediated PFKFB3 repression [18] but also via AMPK-dependent pathways [49]. Vice versa, it was reported that OF upregulates glycolysis in ECs [50,51]. LSS or pulsatile shear stress (PSS) mediated downregulation of glycolysis was found to be regulated to a large extent by KLF4 and AMPK. Initially, PSS induces the transcription of glucokinase regulatory protein (GCKR) via KLF4. Furthermore, pulsatile flow (PF) activates AMPK, which subsequently phosphorylates GCKR at Ser481 to subsequently increase its interaction with hexokinase 1 (HK1). The GCKR-mediated inhibition of HK1 is associated with the suppression of glucose conversion into glucose 6-phosphate, a pivotal stage in glycolysis [49] (Figure 2, left). Further investigation of this pathway in mice indicated that AMPKα2 is the responsible AMPK subunit for the PSS-induced downregulation of glycolysis [49]. In contrast, an increased activation of AMPKα1 in ECs after exposure to DF was shown to result in an increase in glycolysis via mediating an upregulation of HIF1a expression. The latter results, subsequently, in increased transcription of glycolytic genes, such as SLC2A1 and PFKFB3 [52] (Figure 2, right). Interestingly, the authors furthermore reported that inhibiting AMPKα1 under DF decreases glycolytic rates, which resulted in decreased EC viability [52]. These findings demonstrate that discrete AMPK subunits are selectively activated by LSS or DSS, eliciting different physiological responses.

An alternative pathway was proposed in the context of hyperglycemia, where a shear stress/AMPK/miR-181b-mediated improvement of endothelial dysfunction was reported in a rodent model for diabetes [53]. miR-181b is a noncoding microRNA, the expression of which correlates with improved endothelial function in diabetes type 2 patients [54] and which was upregulated in mice exposed to chronic exercise. The upregulation was induced by AMPK, hypothetically in response to the increased shear stress triggered by increased blood flow during exercise. This was furthermore supported by in vitro findings, where LSS exposure of HUVECs activated the AMPK/miR-181b axis [53].

In sum, these results indicate that shear stress-mediated adaptations in glycolysis are crucial for EC physiology and that AMPK is a central link between various stimuli and metabolic adaptions. Generally, in a quiescent monolayer under laminar flow, EC metabolism is downregulated while it is increased under DF, where it potentially serves as a protective adaptation. The differences between AMPK-mediated downregulation of glycolysis in LSS and upregulation of glycolysis in DSS could most likely be explained by the activation of a distinct AMPK subunit by these two distinct mechanical stimuli.

### 3.3. AMPK-Mediated Regulation of Fatty Acid Metabolism

The primary ATP-generating pathway in ECs is glycolysis, although FA metabolism also significantly contributes to EC energy supply. Moreover, the regulated transport of FAs through ECs is crucial for supporting tissue and organ functions [55]. Hydroxy-methylglutaryl coenzyme A reductase (HCR) is both a rate-limiting enzyme for cholesterol synthesis and a substrate of AMPK. It was shown that LSS-mediated AMPK activation leads to HCR phosphorylation, thereby inhibiting its enzymatic activity. This inhibition was mediated synergistically via AMPK-mediated phosphorylation of HCR and by AMPK-mediated, phosphorylation-induced degradation of the transcription factor forkhead box O1a (FoxO1a), which resulted in decreased mRNA levels of HCR [56]. Compelling evidence suggestive of a shear-independent role of AMPK in FA metabolism underscores the potential significance of shear stress-induced AMPK activation in regulating this physiological process. As such, previous work showed that the pathogenic lipid accumulation in ECs could be attenuated by bradykinin-mediated AMPK activation in bovine aortic endothelial cells (BAECs), an effect that was dependent on CaMKK as an upstream kinase of AMPK [57]. This suggests that shear stress-mediated AMPK activation could elicit comparable regulatory effects on ECs.

### 3.4. AMPK-Mediated Regulation of Metabolism via FoxO1

It was shown that shear stress activates AMPK, which phosphorylates the transcription factor FoxO1a, leading to its degradation [56,58]. FoxO1a was previously identified as a gatekeeper of EC metabolism, and its deletion was reported to induce EC proliferation and sprouting in murine ECs, while its forced expression in mice was shown to correlate with decreased glycolytic rates and mitochondrial respiration [59]. Furthermore, HUVECs transduced with a constitutively active FoxO1 mutant (AdFOXO1A3) were reported to adapt a quiescent phenotype, in line with previous data that associate an increased FoxO1 activity with decreased metabolism and quiescence in ECs [60]. Moreover, it was shown that shear stress-mediated activation of AMPK in HUVECs induces Klf2 expression and FoxO1 degradation, which resulted in a downregulation of angiopoietin-2 (Ang-2) [58,61], a cytokine released by ECs during inflammation [62]. These results appear counterintuitive, as regions of LF are characterized by a metabolic quiescent state in ECs.

Based on the abovementioned overexpression studies ([59,60]), FoxO1a expression would be expected to be high in LSS regions to repress proliferation and promote quiescence. However, the two studies reporting shear stress/AMPK-mediated downregulation of FoxO1a in ECs and AMPK/FoxO1-mediated downregulation of the pro-inflammatory cytokine Ang-2 do not support this hypothesis. Instead, these findings indicate that FoxO1 is downregulated in regions of high shear stress, which is further reinforced by prior research demonstrating shear stress-induced activation of Akt via platelet endothelial cell adhesion molecule-1 (PECAM-1) [63], a well-known negative regulator of FoxOs [64]. These apparent discrepancies are yet to be fully understood; however, we hypothesize that the different outcomes can be explained by the different experimental approaches. Forced expression induces significant changes in FoxO activity, potentially leading to different outcomes compared to experiments investigating the impact of shear stress on endogenous FoxO1 activity, wherein FoxO1 activity may be finely adjusted in response to mechanical cues.

## 4. eNOS and NO Production Induced by Shear Stress-Activated AMPK

eNOS (endothelial NO synthase) is an enzyme that produces nitric oxide (NO) in endothelial cells. As such, it is an important regulator of vascular tone and angiogenesis and has atheroprotective and anti-inflammatory effects [65,66]. Moreover, NO is a vasodilator, thereby playing a crucial role in regulating vascular constriction and relaxation [67]. Lastly, decreased NO production is associated with a variety of CVDs, for example, hypertension or atherosclerosis [68,69]. To our knowledge, the earliest study that investigated AMPK in the context of shear activation was conducted by Ingrid Fleming et al. in 2005 [63], who examined the involvement of the EC-specific cell adhesion molecule PECAM-1 in transmitting mechanical stimuli. Prior research has shown that shear stress induces the activation of eNOS and subsequent NO synthesis in ECs [70], and further studies reported that AMPK can activate eNOS via phosphorylation [71]. Expanding on these findings, subsequent studies explored the potential mechanosensitive role of AMPK in HUVECs downstream of PECAM-1. Following the in vitro application of LSS to HUVECs, the authors observed an increased phosphorylation of AMPK at Thr172, first describing AMPK being activated by shear stress. However, this effect was found to be independent of PECAM-1 signaling. Shortly after, it was shown that the activity of the upstream kinase LKB1 increased after BAECs were exposed to LSS and that LKB1 immunoprecipitated with GST-AMPK with resulting eNOS phosphorylation, suggesting a shear-responsive LKB1-AMPK-eNOS pathway in ECs [72].

Further investigations revealed that shear stress-activated AMPK phosphorylates eNOS at Ser633 and Ser1177 [45,71]. A synergistic role for AMPK and SIRT1 (Sirtuin1), a key regulator of stress responses and energy homeostasis, in activating eNOS upon exposure to LF, but not OF, was demonstrated in HUVECs. As such, AMPK-mediated eNOS phosphorylation induces SIRT1-mediated eNOS deacetylation, which leads to enhanced NO production and stimulation of an atheroprotective phenotype in LF conditions [73]. An additional pathway of flow-mediated eNOS activation via MAGI1 (MAGUK with inverted domain structure-1) was reported in 2019 [74]. This study shows that fluid shear stress induces MAGI1 transcription, initiating AMPK and protein kinase A (PKA) activation, which synergistically activates eNOS [74]. Finally, an additional AMPK-dependent, KLF2-mediated eNOS activation pathway induced by PSS or LSS was identified. The authors reported that shear stress activated AMPK in HUVECs and observed downstream phosphorylation of ERK5 and MEF2, which subsequently induced KLF2-mediated eNOS upregulation, likely by increasing its expression. This demonstrates a novel AMPK/ERK5/MEF2/KLF2 pathway in shear-dependent eNOS activation [61]. Altogether, these data show that AMPK is at the center of mechanosensing pathways that activate eNOS upon LF, thereby increasing NO bioavailability and promoting endothelial function in regions exposed to LF (graphical summary: Figure 3).

## 5. AMPK- and Shear Stress-Mediated Endothelial Glycocalyx Impairment

The endothelial glycocalyx (EGX) is a layer of polysaccharides located on the apical, blood-flow-facing side of ECs. A major component of the EGX is hyaluronic acid (HA), and hyaluronidase 2 (HYAL2) is a catabolizing enzyme for HA [75]. The EGX plays a role in the maintenance of the EC barrier function by regulating molecular transport and mechanosensation [76,77]. The latter is closely linked to shear stress-mediated effects on AMPK and subsequent effects in ECs. As such, it was shown that DF activates HYAL2 and leads to EGX degradation. Interestingly, it was reported that this HYAL2-mediated EGX degradation was the result of a low shear stress-mediated reduction in LKB1 activity, which resulted in a decrease in AMPK activity and increased HYAL2 activity via p47^phox^ [78].

Furthermore, NADPH oxidase was increased in activity upon low shear stress-mediated downregulation of the LKB1/AMPK axis, culminating in a LKB1/AMPK/NADPH oxidase-dependent pathway in ECs that regulates EGX maintenance [79]. The authors furthermore showed that pharmacological activation of AMPK with 5-Aminoimidazole-4-carboxamide1-β-D-ribofuranoside (AICAR) resulted in decreased HYAL2 activity under low shear stress [79], which suggests that the atheroprotective phenotype of ECs in regions of high laminar shear stress is in part regulated via AMPK-mediated downregulation of HYAL2, and the subsequent destabilization of the EGX. In a follow-up study on this pathway, the authors furthermore found that treatment of ECs with berberine, a plant-derived alkaloid used in traditional medicine [80], counteracts low shear stress-induced EGX degradation by upregulating AMPK phosphorylation and downstream LKB1/AMPK/NADPH oxidase pathway alteration to decrease HYAL2 activity. Moreover, berberine was shown to increase the expression of hyaluronan synthase-2 (HAS2) via AMPK, leading to an increased HA synthesis, but the exact mechanism of action has yet to be identified [78].

Another study investigating the role of AMPK in low shear stress-mediated EGX degradation identified Na+-H+ exchanger (NHE)1 as an AMPK substrate, with increased activity upon low shear stress-mediated reduction in AMPK activity [81]. NHE1 catalyzes ion exchange to modify the intracellular PH, which was shown to be an activating stimulus of HYAL2, presumably synergistic to the previously discussed role of p47^phox^ regulation downstream of AMPK. The most recent study investigating the role of AMPK in EGX degradation during sepsis uncovered an additional mode of action, dependent on heparan sulfate (HS)-mediated activation of AMPK [62]. HS is part of the EGX and mechanosensor in ECs [82]. In the context of sepsis, the authors found a degradation of HS in the EGX due to enzymatic cleavage. In vitro degradation of HS in human primary lung microvascular endothelial cells (HLMVECs) was shown to result in decreased AMPK activation by shear stress, which results in decreased FoxO1 phosphorylation and downstream upregulation of Ang2 [62]. This study highlights the role of EGX in mechanotransduction and suggests HS as a novel mechanosensor upstream of AMPK activation and Ang2 expression in ECs. Together, these data show that low shear stress promotes EGX degradation via reduced AMPK activation and subsequent activation of catabolic processes. High shear stress-mediated AMPK phosphorylation thereby plays a key role in maintaining the EGX, and a loss of AMPK activation promotes EGX catabolism, a hallmark of endothelial dysfunction.

## 6. AMPK and Endothelial Autophagy

Macroautophagy (the canonical autophagic pathway, referred to as autophagy hereafter) is a cellular process to clear cells from obsolete proteins and organelles and recycle these to provide metabolites to ensure cell survival and maintenance [83,84]. Importantly, defective autophagy in ECs has been linked to vascular dysfunction [85]. The role of shear stress-induced, AMPK-mediated autophagy has not been studied extensively, but it is well known that AMPK is a key positive regulator of autophagy, counteracted by the mTOR pathway [86]. In addition to the numerous studies outlined in earlier sections of this review, which suggest that shear stress regulates AMPK activity, this supports the idea that shear stress-mediated AMPK signaling plays a crucial role in regulating autophagy. In fact, shear stress-induced AMPK-mediated autophagy has been demonstrated by Vion et al. ([87]), who showed that HUVECS display an increase in autophagy upon exposure to high shear stress and decreased autophagic flux when exposed to low shear stress. The decrease in autophagy under low shear stress was found to result from a shear stress-dependent AMPKα inhibition, and mTOR activation, leading to an atherogenic environment in ECs [87].

Additional evidence supporting the involvement of AMPK signaling in the regulation of EC autophagy has previously been presented in the context of stressors other than shear. As such, it was reported that hypoxia activates AMPK and upregulates PTP-PEST protein levels to induce autophagy. PTP-PEST interacts with AMPKα subunits 1 and 2 under normoxia, leading to AMPK dephosphorylation, but this interaction is lost during hypoxia. Loss of PTP-PEST interaction with AMPK results in its phosphorylation, which upregulates autophagy [88]. It was furthermore shown that hypoxia leads to decreased levels of mTOR activation, further promoting EC autophagy under hypoxia [89]. Moreover, several studies reported a shear stress-regulated activation of autophagy without focusing on the involvement of AMPK. For example, it was shown that LF upregulates autophagy to downregulate YAP, a transcription factor associated with pro-inflammatory gene expression in ECs [46]. On the other side, low shear stress was linked to decreased EC autophagy via TET2 downregulation, an enzyme linked to cytosine demethylation and cell differentiation [90]. Together, these studies indicate that shear stress plays a crucial role in EC autophagy regulation, and AMPK is a major regulator of autophagy in ECs. While a direct link between shear-induced AMPK activation and autophagy in ECs has been reported [87], further research is required to show how shear stress modulates autophagy via AMPK.

## 7. AMPK in CVDs

Most reports described above state that AMPK activity increases in regions of high shear stress and decreases in regions of low shear stress. It has been reported several times that atherosclerosis develops preferentially in regions of low shear stress [91,92]. Vion et al. reported that low shear stress-mediated AMPK downregulation of autophagy plays a key part in the onset and progression of atherosclerosis due to (1) establishing a pro-inflammatory EC phenotype following downstream inhibition of KLF2 and ICAM-1 upregulation (Figure 4, left panel), (2) increased EC apoptosis following an increase in the apoptosis regulator p53 and (3) increased EC senescence mediated by elevated levels of p16 [87].

A further aspect of atherosclerosis progression is endothelial-to-mesenchymal transition (EndoMT), a process where ECs progressively change their phenotype toward mesenchymal identity. It was shown that OF promoted EndoMT in ECs via ROS elevation and that forced activation of AMPK (AICAR) and Sirt1 (adenoviral overexpression) resulted in upregulation of endothelial markers vWF, CD31, CDH5, and a decrease in mesenchymal markers CDH2, FSP1, and vimentin under OF [93] (Figure 4, center panel). An additional atheroprotective mechanism leading to shear stress-induced AMPK activation was reported to be mediated via cortactin. PSS activated AMPK to phosphorylate cortactin, which induced Sirt1 activation. This was shown to result in an AMPK/Sirt1 coregulation, where AMPK phosphorylates and Sirt1 deacetylates cortactin. This was shown to induce subcellular translocation of eNOS from lipid to nonlipid raft domains, leading to an increased atheroprotective eNOS activation [94].

Further evidence of the involvement of shear stress-mediated AMPK regulation in CVDs comes from a study investigating the role of microRNA (miR) expression in coronary artery disease [95]. The authors reported that nucleolin (NCL), a multidomain protein that processes miRs, is phosphorylated at Ser-328 by AMPK upon PSS exposure to ECs. This resulted in a decreased expression of miR-93 and miR-484. Conversely, ECs exposed to DSS showed reduced levels of AMPK phosphorylation and subsequent NCL phosphorylation, followed by increased levels of miR-93 and miR-484, an effect that was lost upon AMPK knockdown. The upregulation of miR-93 and miR-484 resulted in a downregulation of KLF2 by miR-93 and a downregulation of eNOS by miR-484 (Figure 4, right panel). Lastly, the authors found that plasma levels of miR-93 and miR-484 are elevated in patients with coronary artery disease (CAD). This suggests that a DSS-induced downregulation of AMPK phosphorylation induces upregulation of specific miRs that are linked to endothelial dysfunction in CAD [95].

In summary, these data show that the central role of AMPK as a mechanosensitive kinase is crucial in promoting LSS-induced endothelial function. Conversely, the downregulation of AMPK phosphorylation by DSS is associated with increased risk for various forms of CVD, highlighting the significance of shear stress-induced mechanoregulation of AMPK in determining endothelial function.

## 8. AMPK as Potential Therapeutic Target

As outlined above, LSS-mediated AMPK activation in ECs correlates with a quiescent, anti-inflammatory, and healthy endothelium. AMPK is, therefore, a promising pharmacological target for reducing CVD. Metformin is an FDA-approved antidiabetic drug that is widely used for the treatment of diabetes type 2 [96]. Furthermore, it was found to be an AMPK agonist [97]. In the context of diabetes-promoted atherosclerosis, metformin was shown to reduce atherosclerotic lesions in diabetic ApoE^−/−^ mice, and this effect was lost in ApoE^−/−^/AMPKα2^−/−^ mice [98]. Further investigation into the mechanism of action revealed that metformin activates AMPK in HUVECs, resulting in a reduced expression of dynamin-related protein 1 (Drp-1) and a subsequent decrease in high glucose-induced mitochondrial fission. This was shown to result in a reduction in oxidative stress, increased endothelial function, and attenuation of atherosclerotic lesion development [98].

A further study on metformin-induced AMPK activation reported a reduction in OF-mediated EndoMT and oxidative stress in HUVECs upon treatment [93]. It was furthermore shown that metformin-induced AMPK activation reduces hypoxia-mediated endothelial dysfunction in human cardiac microvascular endothelial cells (HCMECs) [99]. Finally, metformin-induced AMPK activation was found to attenuate excessive pro-inflammatory EC responses in sepsis in endotoxic mice via AMPK-regulated HDAC5 phosphorylation and KLF2 upregulation [100].

Berberine is a plant-derived alkaloid, which is widely used to treat gastrointestinal disorders and reduce CVD [101,102]. Several studies reported that berberine activates AMPK [103,104]. In the context of endothelial dysfunction, it was shown that TNFα-mediated upregulation of pro-inflammatory gene expression and nuclear factor (NF)-κB activation in HAECs was attenuated upon treatment with berberine via berberine-induced AMPK activation [105]. Endothelium-dependent vasocontractions (EDVs) represent an additional form of endothelial dysfunction. EDVs are observed upon elevated levels of oxidative stress and cyclooxygenase (COX)-derived prostanoids during hypertension [106]. Berberine was shown to reduce EDVs in carotid arteries of spontaneous hypertensive rats (SHR) by attenuating ROS and COX-2 expression. Interestingly, the mechanism of action was reported to include berberine-mediated AMPK activation and subsequent endoplasmic reticulum stress inhibition [107]. Lastly, berberine was reported to rescue palmitate-mediated decrease in NO availability in HUVECs via AMPK phosphorylation and downstream eNOS activation [108].

Another class of compounds known to activate AMPK are statins [109,110]. In an in vivo model for ischemia-induced angiogenesis in mice, pravastatin was reported to increase AMPK phosphorylation, which mediated an improvement in angiogenesis upon ischemic injury, as well as an increased capillary density compared to untreated controls [111]. In vitro follow-up on these findings in HUVECs revealed that pravastatin induced AMPK-dependent eNOS activation, while inhibition of eNOS halted the effects of pravastatin on angiogenesis in mice [111]. This suggests that pravastatin can increase the phosphorylation of AMPK upon ischemic injury and subsequently increase eNOS activity to promote NO bioavailability and angiogenesis. Finally, atorvastatin was reported to increase AMPK phosphorylation in HUVECs and induce AMPK downstream activation of acetyl-CoA carboxylase and eNOS. Furthermore, mice treated with atorvastatin had elevated levels of AMPK phosphorylation and showed increased eNOS phosphorylation in the aorta and myocardium [112]. These results suggest that statins are promising therapeutic agents by activating AMPK to increase NO bioavailability and attenuate endothelial dysfunction.

## 9. Conclusions

Shear stress sensing and metabolic adaptions are two features of EC physiology that have been studied extensively in separate contexts. Shear sensing caught much attention in the context of atherosclerotic plaque development due to the clearly nonrandom locations of plaques near aortic bifurcations where flow is turbulent. EC metabolic adaptions have been mainly studied in the context of angiogenesis and neovascularization, where metabolic adaptions regulate the switch between EC quiescence and proliferation. Interestingly, AMPK stands at the center of these two features, connecting shear sensing and metabolic adaptions in ECs. As summarized in the present review, most studies report that LF induces phosphorylation of AMPK, and TF induces dephosphorylation. However, one study reports DSS-induced phosphorylation of a specific AMPK subunit in the context of glycolysis ([52]), postulating a protective, active mechanism rather than a result of low shear stress exposure. This further highlights the importance of focusing on selective AMPK subunit activation in future studies.

To date, limited knowledge exists regarding how ECs sense mechanical stimuli on their surface and subsequently translate these signals to modulate AMPK activity. The identification of additional mechanosensors on the cell surface that communicate with AMPK could offer valuable targets for drug development to address AMPK dysregulation in CVD. Recently, PLXND1 was found to be activated by mechanical stress and identified as a crucial mechanosensor in ECs that facilitates alignment with flow and upregulation of anti-inflammatory genes under LSS. Furthermore, loss of PLXND1 in OSS regions resulted in an attenuated upregulation of inflammatory markers [113]. Although the downstream signaling cascade of PLXND1 has yet to be fully elucidated, its activation and downstream effects in response to LSS exposure are similar to those of flow-mediated AMPK activation. A plausible explanation is that activation of AMPK is a downstream effect of PLXND1 activation, suggesting a PLXND1/AMPK signaling cascade as an interesting angle for compound studies. Together, these findings indicate that PLXND1 is a mechanosensor in ECs, which stimulates the expression of anti-inflammatory genes under LSS and inhibits the expression of pro-inflammatory genes under OSS. The latter findings indicate that PLXND1 activation has differential effects under LSS and OSS, implying either differential AMPK subunit activation after exposure to different stimuli or separated signaling cascades for distinct shear stress dynamics altogether.

In addition to focusing on mechanosensitive signaling cascades, AMPK’s pivotal role as a central kinase in EC physiology positions it as a prime candidate for direct drug targeting. Previous reports showed that drugs that are already employed in human medicine, such as metformin or berberine, can directly target AMPK. Progress in the research of AMPK as a therapeutic target will likely be made using improved in vitro models, such as vessel-on-chip models, that incorporate human ECs and allow the modulation of mechanistic parameters, including shear stress, cyclic stretch, and multicellular crosstalk.

## Figures and Tables

**Figure 1 ijms-25-06047-f001:**
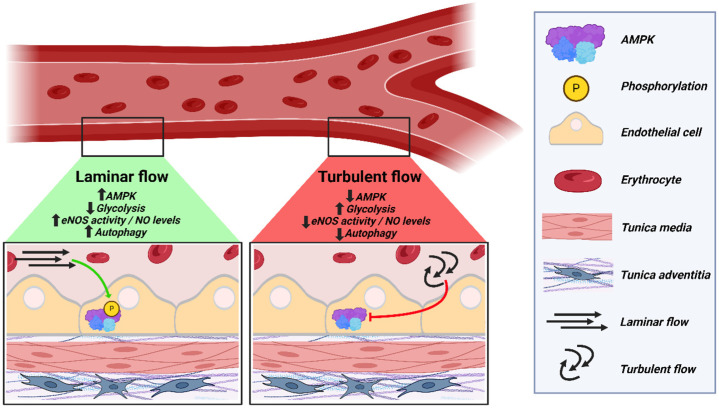
Graphical summary: Overview of AMPK regulation in the endothelium focusing on EC mechanosensation. Schematic cross-section outlining endothelial barrier (tunica intima), tunica media, and tunica adventitia of the human aorta. In straight areas of the vascular system, such as in the linear part of the aortic branch, blood flow is laminar and shear stress is high (**left panel**). This triggers increased AMPK phosphorylation levels in ECs, leading to its activation. This results in reduced glycolysis, increased eNOS activity, NO levels, and autophagic flux. Due to bifurcations or a bent geometry, such as in the curved area of the aortic arch, the blood flow is turbulent and exerts low shear stress (**right panel**). Consequently, AMPK phosphorylation is reduced in ECs located in regions characterized by this geometry. As a result, EC glycolysis increases, and eNOS activity, NO levels, and autophagic flux decrease. These alterations of ECs exposed to turbulent flow prime the area for developing CVDs such as atherosclerosis.

**Figure 2 ijms-25-06047-f002:**
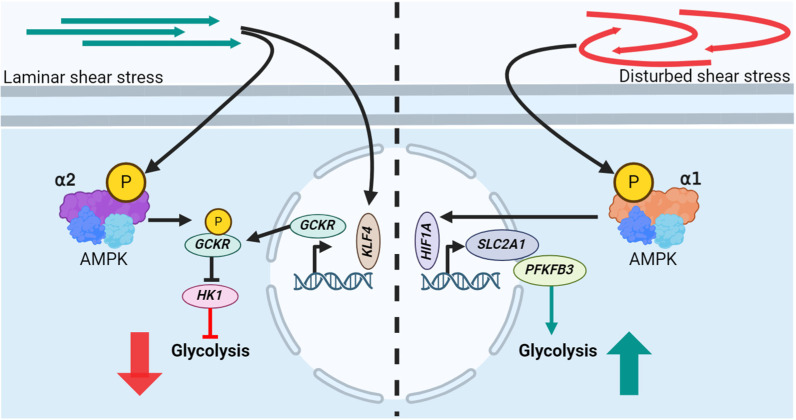
Distinct AMPK subunits are activated by different flow patterns, leading to differing EC glycolysis rates. **Left**: LSS activation of AMPKα2 leads to phosphorylation of glucokinase regulatory protein (GCKR) and simultaneously increases GCKR expression via KLF4. This leads to an increased amount of phosphorylated GCKR, mediating an inhibition of hexokinase 1 (HK1) and subsequent inhibition of glycolysis. **Right**: DSS-mediated phosphorylation of AMPKα1 results in an upregulation of HIF1a expression, which mediates an increased transcription of the glycolytic genes SLC2A1 and PFKFB3, thereby increasing glycolytic rates.

**Figure 3 ijms-25-06047-f003:**
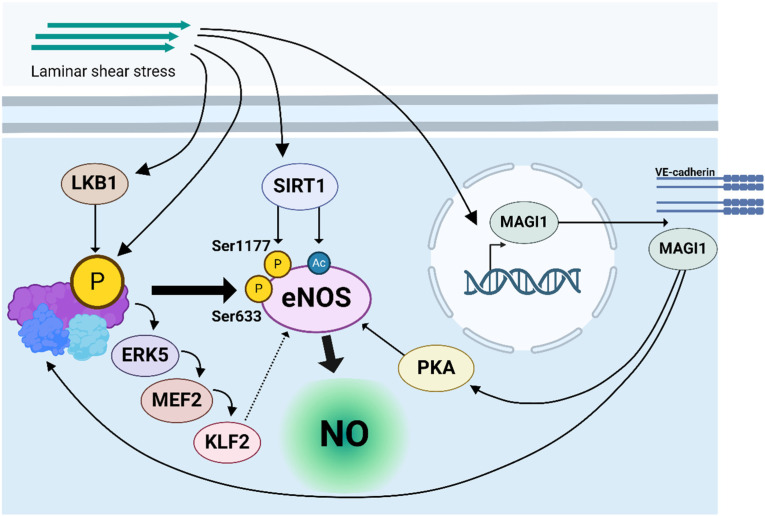
Overview of pathways that increase NO levels in ECs upon LSS-induced AMPK activation. (1) LSS induces AMPK phosphorylation via LKB1, leading to increased eNOS phosphorylation at Ser633 and Ser1177, thereby enhancing its activity to generate NO. (2) LSS leads to AMPK phosphorylation simultaneously to Sirtuin1 (SIRT1) activation, leading to an AMPK-mediated phosphorylation of eNOS and SIRT1-mediated eNOS deacetylation, which further enhances eNOS-mediated NO production. (3) LSS induces MAGI1 (MAGUK with inverted domain structure-1) transcription, which is localized at VE-cadherin junction complexes and subsequently induces AMPK phosphorylation and protein kinase A (PKA) activation, which in turn synergistically enhances eNOS activity. (4) LSS induces AMPK phosphorylation and a downstream cascade of ERK5, MEF2, and KLF2 activation. KLF2 subsequently was shown to increase active eNOS levels, likely via increased transcription rates, leading to an increase in NO levels.

**Figure 4 ijms-25-06047-f004:**
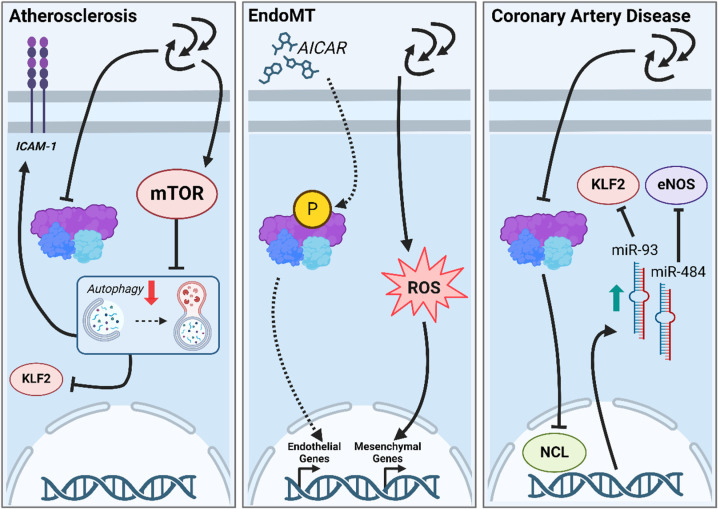
Overview of the role of shear stress-mediated AMPK signaling in cardiovascular diseases. **Left**: Atherosclerosis. TF reduces AMPK activity via decreased phosphorylation and activates the mammalian target of rapamycin (mTOR) in ECs. Decreased AMPK activity and increased mTOR activity inhibit the autophagic flux. Impaired autophagy results in an inhibition of KLF2, increased inflammation (i.e., ICAM-1 expression), apoptosis, and senescence, thereby creating a proatherosclerotic EC phenotype. **Center**: EndoMT. TF generates ROS in ECs, leading to an upregulation of mesenchymal gene transcription, thus driving EndoMT. Pharmacological activation of AMPK via AICAR mimicking LSS leads to an upregulation of endothelial cells, thereby promoting a healthy endothelium and counteracting TF-induced EndoMT. **Right**: Increased levels of miR-93 and miR-484 were found in coronary artery disease patients. TF mediated reduction in AMPK phosphorylation levels in ECs was reported to inhibit nucleolin (NCL) phosphorylation, which leads to increased levels of micro RNAs miR-93 and miR-484. miR-93 subsequently inhibits the LSS-associated transcription factor KLF2, and miR-484, the NO-generating enzyme eNOS.

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
