# Peer review of "Shear Stress-Induced AMP-Activated Protein Kinase Modulation in Endothelial Cells: Its Role in Metabolic Adaptions and Cardiovascular Disease"

_ijms, 2024, doi:10.3390/ijms25116047_

Round 1

Reviewer 1 Report

Comments and Suggestions for Authors

This is a review article summarizing AMPK in endothelial cells from a broad perspective, and there are many things readers can learn from this paper.

However, there are instances of inappropriate wording and referencing throughout.
For example, in lines 87-89, it states: "Despite their direct exposure to oxygen in the bloodstream, ECs exhibit a notably low mitochondrial content [19]."
The original text from the referenced review article, Reference 19, states: "In endothelial cells, mitochondria comprise less than 5% of total cellular volume (Fig. 1) compared to around 28% in hepatocytes [13]."
Original paper cited here (J Cell Biol. 1977 Feb;72(2):441-55.) showed that endothelial cells are not notably low in mitochondria, but rather that hepatocytes are more mitochondria-rich than other cells. This paper does not lead to the conclusion that endothelial cells have low mitochondria.

In this manner, relying on review articles for specific phenomena or reports can often distort facts.
High-quality review articles always cite original sources at crucial points, and it is important to ensure this in the present manuscript as well.

From line 139 to 141, I really did not understand whether the slash "/" was used to indicate a comparison or to mean "and." It needs to be revised to be written in a way that clarifies its intent.

Author Response

Reply to reviewer comments IJMS-2962890.

We highly appreciate the time taken by all reviewers to read and comment our manuscript. Below we respond to the individual points raised. Changes in the text of the manuscript in response to the various comments are marked in yellow and shown through ‘track changes’.

Reviewer 1

This is a review article summarizing AMPK in endothelial cells from a broad perspective, and there are many things readers can learn from this paper. However, there are instances of inappropriate wording and referencing throughout.

For example, in lines 87-89, it states: "Despite their direct exposure to oxygen in the bloodstream, ECs exhibit a notably low mitochondrial content [19]."

The original text from the referenced review article, Reference 19, states: "In endothelial cells,

mitochondria comprise less than 5% of total cellular volume (Fig. 1) compared to around 28% in

hepatocytes [13]."

Original paper cited here (J Cell Biol. 1977 Feb;72(2):441-55.) showed that endothelial cells are not notably low in mitochondria, but rather that hepatocytes are more mitochondria-rich than other cells. This paper does not lead to the conclusion that endothelial cells have low mitochondria. In this manner, relying on review articles for specific phenomena or reports can often distort facts.

High-quality review articles always cite original sources at crucial points, and it is important to

ensure this in the present manuscript as well.

We thank the reviewer for the constructive feedback. Indeed, the respective section of our review (Section “Introduction”, paragraph 11) could lead to a false conclusion and we addressed this point by changing the wording to the following:

 “Despite their direct exposure to oxygen in the bloodstream, ECs exhibit a relatively low mitochondrial content (2-6%) [19], compared to other cells such as hepatocytes (28%) [20] or cardiomyocytes (~30%) [21]. It was furthermore shown that ECs depend on anaerobic glycolysis as the primary mechanism for adenosine triphosphate (ATP) production [22, 23].”

In this way, we hope to more clearly pointed out the intended meaning, stating that endothelial cells have a relatively low mitochondrial count compared to other cell types that rather rely on oxidative phosphorylation for energy production. We furthermore included references to primary research papers to further strengthen this point.

From line 139 to 141, I really did not understand whether the slash "/" was used to indicate a

comparison or to mean "and." It needs to be revised to be written in a way that clarifies its intent.

Thank you for pointing this out. We hope to have clarified this point by changing the slash to a “and” as we aim to convey the message that AMPK is activated if the ratio shifts from high ATP levels and low AMP and ADP levels, towards low ATP levels and high AMP and ADP levels. We therefore have rewritten this passage as follows:

“The canonical activation of AMPK acts via sensing ATP to AMP and ADP (adenosine diphosphate) ratios, which are shifted during metabolic stress towards higher ADP and AMP levels.” (Section “Structure and activation of AMPK”, paragraph 3)

Reviewer 2 Report

Comments and Suggestions for Authors

This manuscript by Philipp C. Hauger and Peter L. Hordijk reviewed role of shear stress-induced endothelial AMPK modulation and its role in metabolic adaptions and CVD. Authors concisely assembled the literature to understand the how shear stress in ECs activates AMPK and its proteins, and their role on metabolic adaption as the central mechanosensing protein and how AMPK dysregulation involved in CVD. This is a simple review and useful in the context of shear-stress and AMPK role. However, there are multiple sections which authors must improve before it get accepted for publication.

1) Title sounds very much suited for what this review is aimed.

2) Abstract: There are few lines, very much unclear to me. Eg. 7-8 " mediates the regulated transport". please provide abbreviation for AMP. The abstract is poor, please improve.

3) Introduction: The plotting of the statements, and the reading flow is greatly disturbed throughout the manuscript. Please improve the it. A graphical illustration is useful for the Line 25-34. The description of physiological forces in the vasculature is concise and informative.

4) Fig 1-4 are informative. Reg. Fig 4: A clarity is required, whether the right panel is accurate or, vessel obstruction due to the claudication or plaque deposition is the major cause of turbulent flow, which may more suitable in CVD setting. Please improve this part.

Overall, the reading flow is not enjoyable, therefore this manuscript needs extensive English editing before it get accepted. 

Author Response

Reviewer 2

This manuscript by Philipp C. Hauger and Peter L. Hordijk reviewed role of shear stress-induced

endothelial AMPK modulation and its role in metabolic adaptions and CVD. … However, there are multiple sections which authors must improve before it get accepted for publication.

  • Title sounds very much suited for what this review is aimed.

We appreciate this remark of the reviewer

2) Abstract: There are few lines, very much unclear to me. Eg. 7-8 " mediates the regulated

transport". please provide abbreviation for AMP. The abstract is poor, please improve.

We addressed the lack of clarity in line 7-8 (Section “Abstract”) by changing the text to the following:

Endothelial cells (ECs) line the inner surface of all blood vessels and form a barrier which facilitates the controlled transfer of nutrients and oxygen from the circulatory system to surrounding tissues.”

We furthermore screened and adapted the text of the abstract and hope that it is improved as compared to the original version.

Furthermore, an abbreviation for adenosine monophosphate (AMP) is in the “List of abbreviations” (page 14) and in the text at the first occurrence of the term (Section: “Introduction”, paragraph 13).

3) Introduction: The plotting of the statements, and the reading flow is greatly disturbed

throughout the manuscript. Please improve the it.

We have screened the introduction as well as the rest of the manuscript to optimize the text.

A graphical illustration is useful for the Line 25-34.

Thank you for pointing this out. A graphical summary for lines 25-34 is given in figure 4, but due to formatting reasons it was placed at the end of the manuscript. To improve clarification, we added a reference to this figure the introduction (Section: “Introduction”, paragraph 1):

The adventitia comprises fibroblasts and extracellular matrix that provide structural support to the vessels, as well as small vessels (vasa vasorum) and nerve endings [3] (graphical summary: figure 4).”

The description of physiological forces in the vasculature is concise and informative.
We appreciate this remark of the reviewer

4) Fig 1-4 are informative. Reg. Fig 4: A clarity is required, whether the right panel is accurate or,

vessel obstruction due to the claudication or plaque deposition is the major cause of turbulent flow, which may more suitable in CVD setting. Please improve this part.

Thank you for this feedback – Indeed, the figure explanation might lead to a wrong conclusion. We clarified it by changing the wording of the legend for figure 4:

Due to bifurcations or a bent geometry such as in the curved area of the aortic arch, the blood flow is turbulent and exerts low shear stress (right panel). Consequently, AMPK phosphorylation is reduced in ECs located in regions characterized by this geometry. As a result, EC glycolysis increases and eNOS activity, NO levels and autophagic flux decrease. These alterations of ECs ex-posed to turbulent flow prime the area for developing CVDs such as atherosclerosis.

We try to emphasize the point that the straight geometries of vessels (especially larger vessels, where blood volume flow is high) leads to a laminar and high shear stress, which is correlated with AMPK activation and a subsequent atheroprotective EC phenotype. In contrast, areas where the geometry of the blood vessel is bent, such as near bifurcations, the blood flow becomes turbulent (due to the vessel geometry), and this was shown to deactivate AMPK, leading to increased inflammation, oxidative stress and additional effects described in our review. These effects ultimately prime the regions that are exposed to such turbulent flow to developing CVDs such as atherosclerosis.

Overall, the reading flow is not enjoyable, therefore this manuscript needs extensive English editing before it get accepted.

We have screened the manuscript to optimize the text.

Reviewer 3 Report

Comments and Suggestions for Authors

This is an excellent review on the metabolic role of AMPK in endothelial cells, its possible roles in cardiovascular disease and the possibility of AMPK becoming a therapeutic target in the treatment of cardiovascular disease.

The review is excellent across the board and have only minor suggestions.

- AMPK not only plays a metabolic role, but also controls myosin II activation, hence contraction. It'd be useful if the authors can add some insight into the molecular mechanisms controlling contraction and metabolism in endothelial cells and the position of AMPK in this cross-talk.

- In this regard, is there information on the role of AMPK in angiogenesis? EC migration is also an important phenomenon in neoplasia and tissue repair.

- Is AMPK connected to the mechanosensitive pathway dependent of plexin D1 recently described by the Tzima group? If not, speculation to this effect would be interesting and appropriate.

- The authors could include a figure explaining the domains of AMPK and how they relate to the different metabolic and migratory pathways.

Author Response

Reviewer 3

This is an excellent review on the metabolic role of AMPK in endothelial cells, its possible roles in cardiovascular disease and the possibility of AMPK becoming a therapeutic target in the treatment of cardiovascular disease. The review is excellent across the board and have only minor suggestions.

- AMPK not only plays a metabolic role, but also controls myosin II activation, hence contraction.

It'd be useful if the authors can add some insight into the molecular mechanisms controlling

contraction and metabolism in endothelial cells and the position of AMPK in this cross-talk.

- In this regard, is there information on the role of AMPK in angiogenesis? EC migration is also an

important phenomenon in neoplasia and tissue repair.

We agree with this reviewer that these are interesting topics.

We could find only very limited information on the link between AMPK and contractility in endothelial cells. Holmes et al. concluded that ‘that Rho/ROCK and actinomyosin contractility are regulated by AMP/ATP levels independently of AMPK’ (Sci rep 2020 Jun 18;10(1):9926.)  More recently, Holzner et al (J cell sci 2021 Sep 1;134) studied AMPK-mediated phosphorylation of cingulin, a negative regulator of MLC phosphorylation.

Regarding angiogenesis, there is more literature on this topic including a number of recent reviews. The association with angiogenesis relates a.o. to the link between AMPK and HAS (e.g. see the review by Chen and Lozzo J Biol chem 2020 Dec 4;295) which is addressed in our review as well.

However, since we aimed to focus our review on AMPK in mechanosensing by ECs and its role in CVD, we decided not to include a mechanistic overview on its link with contractility, as there is hardly any information available. As to the topic of angiogenesis (which is also more often studied in the context of cancer as compared to CVD), this is already reviewed a number of times in recent years (see also Li et al. Cells  2019 Jul 19;8(7):752) and we deemed it therefore not appropriate to reiterate that topic here as well, also to maintain focus in the current paper. We did, however, include an additional publication that investigated the link between angiogenesis and AMPK phosphorylation upon AMPK mediated upregulation of autophagy in the subsection of our review: ”AMPK and endothelial autophagy”.

Fitting well into the context of the autophagy subsection of our review, the added section is:

“Moreover in the context of angiogenesis, it was demonstrated that heat-denatured ECs in mice enhance autophagy through increased AMPK phosphorylation and reduced Akt and mTOR phosphorylation after the temperature insult. Increased EC autophagy was then shown to promote cell migration and angiogenesis, processes linked to wound recovery of the endothelium [90].”

- Is AMPK connected to the mechanosensitive pathway dependent of plexin D1 recently described by the Tzima group? If not, speculation to this effect would be interesting and appropriate.

We thank the reviewer for their excellent input. Although we did not find (to our best knowledge) that AMPK is linked to PLXND1 yet, it is indeed a promising hypothesis that it is a downstream effector of PLXND1 activation, given the similar effects of PLXND1 activation by shear stress, as compared to AMPK activation with comparable stimuli. We added an appropriate section in our concluding words, hypothesizing a PLXND1/AMPK signaling cascade (Section: “Conclusion”, paragraph 2):

Recently, PLXND1 was found to be activated by mechanical stress and identified as crucial mechanosensor in ECs that facilitates alignment with flow, and upregulation of anti-inflammatory genes under LSS. Furthermore, loss of PLXND1 in OSS regions resulted in an attenuated upregulation of inflammatory markers [113]. Although the downstream signaling cascade of PLXND1 has yet to be fully elucidated, its activation and downstream effects in response to LSS exposure are similar to those of flow-mediated AMPK activation. A plausible explanation is that activation of AMPK is a downstream effect of PLXND1 activation, suggesting a PLXND1/AMPK signaling cascade as an interesting angle for compound studies. Together, these findings indicate that PLXND1 is a mechanosensor in ECs, which stimulates the expression of anti-inflammatory genes under LSS, and inhibits expression of pro-inflammatory genes under OSS. The latter findings indicate that PLXND1 activation has differential effects under LSS and OSS, implication either differential AMPK subunit activation after exposure to different stimuli, or separated signaling cascades for distinct shear stress dynamics altogether.

- The authors could include a figure explaining the domains of AMPK and how they relate to the

different metabolic and migratory pathways.

In the current overview, we did not address the domain structure of AMPK and its relation to its functional roles.  This aspect has been recently expertly reviewed by Steinberg and Hardie in Nature Reviews- MCB (2022). We did address the various subunits of the AMP complex in our current manuscript. However, to the best of our knowledge, there are not many publications out yet, that focus on a different AMPK subunit activation in the context of shear stress activation of AMPK in ECs. However, for regulating glycolytic rates upon shear stress exposure in ECs, it was shown that laminar shear stress can activate another AMPK subunit than disturbed shear stress, and that the activation of different subunits have distinct effects on glycolytic rates in ECs. We have depicted the combined current knowledge about this in figure 2 and outlined in the text (Section: “AMPK mediated regulation of glycolysis”, paragraph 1). To furthermore highlight that this figure tries to emphasize the role of different AMPK subunit activation, we changed the legend of figure 2:

“Figure 2: Distinct AMPK subunits are activated by different flow patterns, leading to differing EC glycolysis rates.”

We furthermore discuss this in our concluding remarks (Section: “Conclusion”, paragraph 1): “[…] This further highlights the importance of focusing on selective AMPK subunit activation in future studies.”

Round 2

Reviewer 1 Report

Comments and Suggestions for Authors

The revised manuscript also improves the readability of the text and the quality of references, making the paper more reader-friendly.